# Effect of positive emotion intervention during late pregnancy on improving colostrum secretion: a randomised control trial protocol

Kechen Xu [1,2] Ying Xie,[3] Xiujun Han,[4] Ying Yu,[4] Suqing Liu,[5] Suliu Wu,[6] Qian Yang [1] Qi Zhang[7]

KX, YX, XH and YY contributed equally.

## ABSTRACT

**Introduction** Though evidence has revealed the beneficial effects of cognitive improvement interventions on breastfeeding, the effect of psychological interventions has rarely been studied. This study aims to test whether promoting a positive emotion intervention, 'Three Good Things' intervention, during the last trimester of pregnancy can enhance early colostrum secretion and breastfeeding behaviours by modulating the hormones associated with lactation (prolactin and insulin-like growth factor I). We will attempt to promote exclusive breastfeeding by using physiological behavioural measures.

**Methods and analysis** This study is designed as a randomised controlled trial conducted in the Women's Hospital School of Medicine at Zhejiang University and the Wuyi First People's Hospital. The participants will be randomly divided into two groups using stratified random grouping: the intervention group will receive 'Three Good Things' intervention, while the control group will write about three things that come to mind first. These interventions will be continued from enrolment until the day of delivery. Maternal blood hormone levels will be tested approaching delivery and the following day after birth. Behavioural information about breastfeeding will be collected 1 week afterwards.

**Ethics and dissemination** The study has been approved by the Ethics Committees of the Women's Hospital School of Medicine at Zhejiang University and the Wuyi First People's Hospital. Results will be disseminated through peer-reviewed journals or international academic conferences.

**Trial registration number** ChiCTR2000038849.

## STRENGTHS AND LIMITATIONS OF THIS STUDY

⇒ A randomised controlled trial of pregnant women will be performed to evaluate the effect of 'Three Good Things' on their postpartum emotions, colostrum secretion and breastfeeding behaviours.

⇒ 'Three Good Things' is a typical psychological intervention used for promoting positive emotions, which has shown significant effects.

⇒ 'Three Good Things' is feasible to carry out by individuals and easy to disseminate in daily routines.

⇒ We will promote breastfeeding-related behaviours in the postpartum hospitalisation period in the setting of a Baby-Friendly hospital, in line with future trends in China.

⇒ The result interpretations on breastfeeding behaviours are limited in postpartum period due to the limits of existing conditions.

For numbered affiliations see end of article.

**Correspondence to**
Qian Yang;
chianyoung@zju.edu.cn

## INTRODUCTION

Breastfeeding is one of the most valid ways for promoting children's health and survival, which is advocated by the WHO and the United Nations Children's Fund (UNICEF).[1] Breast milk, with its balanced nutrition, suitable temperature and economic benefits, is considered the most ideal natural food for infants.[2 3] In addition, breast milk contains a diverse array of microbiota and myriad biologically active components that contribute to the development of infants' immune systems. Thus, compared with formula-fed infants, breastfed children have fewer incidences of disease and lower mortality rates.[4 5] Breastfeeding also has economic benefits for individual families as well as at the national level by improving child development and reducing health costs.[6] As the WHO suggests, infants should be exclusively breastfed (given nothing else but breastmilk) for the first 6 months of life and be allowed to breastfeed to 2 years of age or longer.[7] In 2012, the 56th World Health Assembly set a goal for 2025 to increase exclusive breastfeeding rates in the first 6 months to at least 50%. However, few countries in the world fully meet the recommended standards, according to a recent report by UNICEF and the WHO in collaboration with the Global Breastfeeding Collective, a new initiative to increase global breastfeeding rates.[8] The *Global Breastfeeding Scorecard*, which evaluated 194 nations, found that only about 40% of children are breastfed exclusively within the first 6 months of life,

and only 23 countries exceed the standard of 60% on their exclusive breastfeeding rates.[8]

To promote exclusive breastfeeding in the first 6 months postpartum, a large number of studies have explored the influencing factors and intervention measures. A previous study indicated that a positive antenatal attitude towards breastfeeding was the strongest predictor and was associated with a 20%–30% increase in breastfeeding initiation and maintenance at all time points.[9] Accordingly, the Breastfeeding Support Programme was carried out to improve the attitudes and cognitions about breastfeeding among pregnant women. It was found to be an effective measure to delay the cessation of breastfeeding, increasing both its duration and its exclusivity.[10]

In addition to cognitive factors, psychological status and emotions also play an important part in breastfeeding. Current evidence suggests that anxiety and depression are associated with lower exclusive breastfeeding behaviours.[11–13] Psychological distress resulted in low appetite, weight loss and poor nutritional status, all of which made it difficult to breastfeed.[14] Emotional and traumatic stressors were found to have the greatest impact on breastfeeding outcomes.[15] Of clinical relevance, addressing prenatal and postpartum distress as a part of the implementation of breastfeeding practice interventions could improve breastfeeding rates.[16] Although intervention studies have been focusing on psychological distress, few studies have conducted positive psychology interventions (PPIs) to promote breastfeeding. PPIs can enhance women's mental well-being during the perinatal period.[17] Thus, PPIs may be worth exploring as measures to promote breastfeeding behaviours.[18] The 'Three Good Things', as a PPI instructs participants to write down three good things every night, was proposed by Seligman *et al* in 2005. It helps to develop skills in regulating emotions,[19] which can enhance the experience of positive emotions and alleviate the symptoms of depression.[20] This psychological intervention has been tested in multiple studies and is replicable and easy to implement by individuals. Therefore, this study will use and evaluate the effect of 'Three Good Things' as an intervention to enhance positive emotions and reduce negative emotions among pregnant women.

Breastfeeding behaviours are influenced by lactation capacity, which is reflected in hormone levels. Prolactin (PRL) is a protein hormone secreted by the anterior pituitary gland that can stimulate and maintain lactation. Insulin-like growth factor I (IGF-I) is a kind of polypeptide that can mediate the galactagogue of growth hormone. Previous studies have shown that emotions and the secretion of these two hormones are closely related. Positive emotions can promote the secretion of prolactin, while depression can lead to a prolactin secretion imbalance and stress events and anxiety can also lead to a low secretion level of IGF-I.[21–23] This study aims to use 'Three Good Things' intervention to promote positive emotions among pregnant women during mid-pregnancy and late pregnancy, thus improving the secretion of lactation-related hormones and the initiation and continuation of breastfeeding behaviours.

## METHODS AND ANALYSIS
### Aims
The study is designed as a randomised controlled trial (RCT) in China, to test whether the 'Three Good Things' intervention during late pregnancy can enhance colostrum secretion and related breastfeeding behaviours and effects. We also aim to examine whether the association between prenatal emotions and postnatal breastfeeding behaviours are mediated by lactation-related hormones.

### Hypotheses
Hypothesis 1a: 'Three Good Things' interventions can promote the secretion of PRL and IGF-I by enhancing positive emotions and reducing negative emotions in pregnant women.

Hypothesis 1b: 'Three Good Things' interventions can improve maternal breastfeeding willingness and postpartum in-hospital breastfeeding behaviours by enhancing positive emotions and reducing negative emotions.

Hypothesis 2a: Increased secretion of PRL and IGF-I will predict the promotion of breastfeeding behaviours and effects.

Hypothesis 2b: Positive emotions can indirectly promote breastfeeding behaviours during postpartum hospitalisation through the secretion of PRL and IGF-I, while negative emotions have adverse effects.

### Study design
This is the protocol for a randomised controlled clinical trial. Between October 2020 and June 2021, a pilot study was carried out at the First People's Hospital of Wuyi County in Zhejiang Province. During this time, a total of 96 pregnant women were recruited for the test of the designed survey questionnaire, including the baseline survey and follow-up survey and intervention questionnaire. In January 2022, the formal trial was conducted at Women's Hospital School of Medicine at Zhejiang University. This study will have three phases: $T_0$, baseline; $T_1$, immediate postintervention and $T_2$, follow-up (1 week after delivery). The qualitative and quantitative data will be collected in three phases. $T_1$ is a baseline survey collecting demographic information, data about existing positive and negative emotions and their intention to breastfeed. $T_2$ is the intervention process for the intervention and the control groups, with process variables of emotion and willingness collected every week. $T_3$ is the prenatal and postnatal hospitalisation period when we will collect outcome indicators such as serum hormone levels, positive and negative emotions, intention to breastfeed, breastfeeding behaviour and other indicators. Figure 1 illustrates the overall study design.

### Recruitment
The study will be conducted in the Women's Hospital School of Medicine at Zhejiang University. We will include

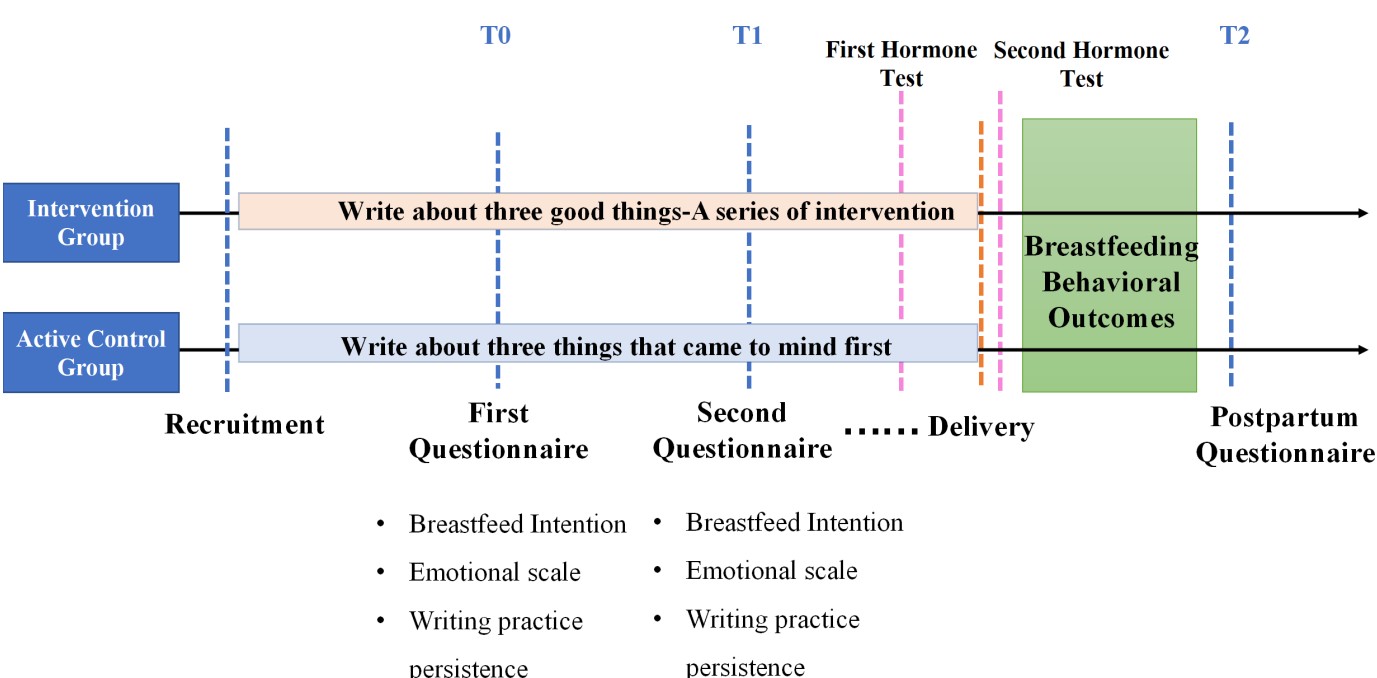

**Figure 1** Flow chart of the breastfeeding intervention project. The participants will be randomly divided into intervention group and control group. In phase $T_0$, participants will complete the first questionnaire, including breastfeed intention, emotional scale and writing practice persistence. In the $T_1$ stage, the participants will complete a series of intervention questionnaires, including breastfeed intention, emotional scale and writing practice persistence. First lactation hormone test will be performed before delivery. The second lactation hormone test will be performed after delivery. Finally, a postpartum questionnaire will be performed about the outcome of lactation behaviour.

primiparous pregnant women at 34–38 gestational weeks. The exclusion criteria are as follows:

► Women with no intention to give birth in a designated hospital.
► Women not giving birth for the first time.
► Women with a disease that prevents breastfeeding (cardiac insufficiency, infectious diseases, neurological or cognitive damage, etc)
► Pregnant women with more than one fetus.

### Sample size
Based on the results of the pilot study, statistical power (1–β) of 80%, and a two-tailed significance level (α) of 0.05, we calculated the targeted sample size with a two-sample t-test allowing unequal variance. Given the breast-feeding behaviour score (LATCH) was 12.72 in the intervention group (SD=2.33) and 12.06 in the control group (SD=1.51), the theoretical sample size was shown as N1=N2 = 141. Considering a dropout rate of 20%, the ultimate sample size needed was estimated at 353 individuals.

### Randomisation and blinding
Medical staff at the hospital's fetal heart monitoring studio will help recruit participants. We will use stratified random grouping to divide the pregnant women into two groups according to the level of support they say they receive (high, average and none) from family members in the oral interview. Baseline data will be collected before the randomisation. Pregnant women received on the same day will be randomly assigned to the intervention group and the control group according to the level of support (high, average and none). KCX will generate the randomisation list, and YX will carry out the randomisation and give the participants information about which group they are allocated to. Our trial will be a single-blinded trial of participants.

### Intervention
At the fetal heart monitoring studio, we will ask the participants to add our Wechat account, and the subsequent communications and relevant questionnaire distribution will be carried out through Wechat chat software. In the $T_1$ stage, participants will complete a series of intervention questionnaires. Participants in the intervention group will be asked to write down three good things that have happened during the day every 2 days, with weekly specific themes. Participants in the control group will only have to write about three things that come to mind first that are related to the same theme. To improve the compliance and interest of the participants, we will set up different instructions and themes for them weekly (table 1). Meanwhile, if a participant forgets to write, we will send out hints if they fail to complete the questionnaires. When the participant is unwilling to take the initiative to fill in the questionnaire, we will ask for reasons and send encouraging messages (eg, about the benefits of participating in the project). For ethical considerations of avoiding excessive disturbance, the participants will be regarded as loss-to-follow-up, after three non-responses.

**Table 1** Instructions for the intervention and control groups

| Week | Topic | Instructions for the intervention group | Instructions for the control group |
|---|---|---|---|
| 1 | My baby | Please recall three good things that happened to you today related to you and your baby, and record them with the reasons why they happened. Here are some examples:<br>1. My baby is moving today, and my baby is growing healthily.<br>2. I saw a cute baby picture today, and my baby may be cuter.<br>3. I bought baby's little clothes today, they are so small and cute. | Please record the first three baby-related events that come to your mind today, and record them and the reasons why they happened. Here are some examples:<br>1. I suddenly thought that it is not easy to have a child.<br>2. I had the maternity check-up today. |
| 2 | Nature and me | Please recall three good things that happened to you today related to nature, and record them with the reasons why they happened. Here are some examples:<br>1. I heard a bird chirping on the road today, is it saying hello to me?<br>2. The small flower we planted is blooming, we have taken good care of it.<br>3. The sky is blue today, and there are traces of airplanes flying over. | Please record the first three nature-related events that come to your mind today, and record them and the reasons why they happened. Here are some examples:<br>1. It's raining today, and the air is wet.<br>2. I watered the flowers at home. |
| 3 | My family | Please recall three good things that happened to you today related to your family, and record them with the reasons why they happened. Here are some examples:<br>1. My husband is very considerate, and I feel very happy.<br>2. My mother and mother-in-law specially cook duck for me, they always love me very much.<br>3. All the people I love and I am expecting the baby together. My friends also told me just take it easy and not to be nervous. | Please record the first three family-related events that come to your mind today, and record them and the reasons why they happened. Here are some examples:<br>1. My husband and I had dinner with our parents today.<br>2. A good friend of mine got married today. |
| 4 | My future | Please recall three good things that happened to you today related to the future, and record them with the reasons why they happened. Here are some examples:<br>1. Today, I saw other people's little kids, this time next year I can have my own baby!<br>2. The weather forecast says the weather will be good, I'm looking forward to sunshine tomorrow.<br>3. Tomorrow my favourite TV series will be updated. | Please record the first three future-related events that come to your mind today, and record them and the reasons why they happened. Here are some examples:<br>1. We're going for a check-up tomorrow.<br>2. Tomorrow my mother-in-law is coming to visit us. |
| 5 | My life | Please recall three good things that happened to you today related to your life, and record them with the reasons why they happened. Here are some examples:<br>1. Today, I listened to my favourite songs, which gave me a lot of inspiration and memories.<br>2. Today, I found a gentle sentence on my phone, which said, 'I was born mediocre, and I was born proud. Let's be a happy ordinary person'.<br>3. The emoticons my friend sent me today were so cute, I thought they were very interesting and funny. | Please record the first three life-related events that come to your mind today, and record them and the reasons why they happened. Here are some examples:<br>1. I watch videos on the phone for an hour today.<br>2. I watched a TV series with my husband this evening. |

## Evaluation and outcomes

Primary outcomes will include exclusive breastfeeding and on-demand feeding during postnatal hospitalisation. Exclusive breastfeeding means that the baby is not given any liquid or solid food other than breast milk. This indicator will be included in the postnatal questionnaire and also be double-checked in the hospital care records of formula. On-demand feeding means breastfeeding according to the baby's needs. Since the infant's early urination reflects breastfeeding, changing diapers 6–8 times a day in the early postpartum period is considered feeding on demand.

Secondary outcome indicators will mainly include other breastfeeding behaviours during postpartum hospitalisation (first postnatal food for babies is breast milk; LATCH breastfeeding score), lactation-related hormone secretion (prolactin and IGF-I) and postpartum breastfeeding intention and postpartum emotion.

## Data collection and follow-up

On the day of enrolment, pregnant women will be asked to complete an enrolment questionnaire providing information about general demographic characteristics and other factors influencing breastfeeding. Meanwhile, if the enrolment questionnaire finds prenatal depression among the participants, we will refer them to the department of psychology. Then, the ecological instantaneous evaluation of both groups will be performed every week to determine the trend of change in intention and emotions until delivery. The measurement of emotion uses the Positive and Negative Affect Scale (PANAS) and 10 positive items for positive emotions proposed by Fredrickson for validation.[24 25] Postpartum questionnaires on breastfeeding behaviour during postnatal hospitalisation will be completed 1 week after delivery. Blood samples will be collected twice, after they are admitted to the hospital approaching delivery and after delivery at 7 to 8am (vaginal delivery: the morning after one night; caesarean

section: the morning after two nights), and evaluated for prolactin and IGF-I to evaluate postpartum lactation ability. The blood draw will be conducted by nurses in the ward.

On recruitment, general demographic variables will be collected: education, occupation, number of permanent household members, total average annual household income and expected date of birth. Other influencing factors will include mode of delivery, self-reported ease of breastfeeding, degree of sleep disturbance, self-reported importance of early exclusive breastfeeding, knowledge of breastfeeding, birth injury and nipple cracking pain scores, degree of hospital support, depression (measured by Edinburgh Postnatal Depression Scale, EPDS), anxiety (measured by Generalised Anxiety Disorder, GAD-7), breastfeeding intentions (measured by Infant Feeding Intentions Scale, IFI), breastfeeding behaviour score (LATCH), resilience (Brief Resilience Scale, BRS) and potential number of traumatic events.[26–31] Generally, Baby-Friendly hospitals will take measures such as early sucking and skin-to-skin contact within 1 hour after delivery, limitation of bottle used, room-in, etc.[32] We will also record and report routine measures taken in this Baby-Friendly hospital.

All questionnaires will be made through the questionnaire star website and distributed via WeChat. The corresponding data are stored in the questionnaire star account. Hormone test results will be recorded and saved manually. We will keep the personal information and data of each participant strictly confidential.

### Data analysis
Descriptive analyses will be performed to show the baseline characteristics of the participants. To assess the intervention effects over time, multivariate analysis of variance of repeated measuring will be taken for the change of positive and negative emotions, intention to breastfeed and resilience trend across the weeks. To test the association among positive/negative emotions, lactation-related hormones and breastfeeding behaviours, we will perform multiple linear regression or logistic regression in between and exploratory mediation analysis. To evaluate the effect of the intervention, we will take a two-sample t-test or Wilcoxon rank-sum test for quantitative data and a $\chi^2$ test for qualitative data.

### Data monitoring
The data monitoring committee is composed of members of the Science and Education Department of the hospital. As the administrative department of the hospital, the Ministry of Science and Education is in charge of scientific research and reviews the integrity of scientific research to ensure the authenticity of data.

### Patient and public involvement
The development of the research question and outcome measures is not influenced by patients' priorities, experiences and preferences. Participants and the public do not involve in the design, recruitment or conduct of the study. We will disseminate the results to study participants through subsequent publications. For this RCT, the burden of the intervention is assessed by the patients themselves through the questionnaire.

### DISCUSSION
Our study has several strengths. First, as distinct from current interventions focusing on knowledge improvement, our study attempts to promote the secretion of lactation-related hormones by elevating positive emotions, then breastfeeding behaviours. Second, most previous interventions promoting breastfeeding were set in the postpartum period. Considering the importance of early breastfeeding experiences, we moved the intervention period to the late stages of pregnancy. Third, earlier mixed educational and supportive interventions are difficult to evaluate and replicate precisely. Our study will use the typical psychological measures, 'Three Good Things', to boost positive emotions, which is feasible for individuals to carry out and can be easier to promote and popularise among pregnant women.

This study also has several limitations. First, due to the restrictions of the testing technology and the intricacies involved in distinguishing between endogenous and exogenous sources, we will not test for oxytocin in the serum, although this has been widely reported to be associated with lactation. However, oxytocin release is related to prolactin levels, which will be included in our test.[33] Second, we limit the outcome measurement of breastfeeding-related behaviours to the postpartum hospitalisation period, without following a 6 month or longer period. Nevertheless, exclusive breastfeeding in the early postpartum period (1 hour and within a few days after delivery) has been positively correlated with the duration of breastfeeding after delivery.

In conclusion, this study will test the effect of 'Three Good Things' intervention both on the emotions of women during pregnancy, and also on the secretion of colostrum and breastfeeding behaviours as well. This study will also help us to further identify the physiological mechanisms of the association between emotions and breastfeeding behaviours.

### ETHICS
This study involves human participants. The study has been approved by the Ethics Committees of the Women's Hospital School of Medicine at Zhejiang University (20190088) and the Wuyi First People's Hospital (20210001). Participants gave informed consent to participate in the study before taking part.

### Author affiliations
[1]School of Public Health, and Department of Geriatrics of the Fourth Affiliated Hospital, Zhejiang University School of Medicine, Hangzhou, China
[2]Clinical Laboratory Center, Wuyi First People's Hospital, Wuyi, China

³School of Health Policy and Management, Chinese Academy of Medical Sciences and Peking Union Medical College, Beijing, China

⁴Department of Obstetrics, Women's Hospital School of Medicine Zhejiang University, Hangzhou, Zhejiang, China

⁵Department of Gynecology, Wuyi First People's Hospital, Jinhua, Zhejiang, China

⁶Department of Science and Education, Wuyi First People's Hospital, Jinhua, China

⁷Community and Environmental Health, Old Dominion University, Norfolk, Virginia, USA

**Acknowledgements** We appreciate the support of the clinical staff of Zhejiang University School of Medicine and Wuyi First People's Hospital, including Xinyi Zhou, Suqing Liu, Xiaohong Qin, Qin Geng and Di Gu.

**Contributors** KX, YX, XH and YY contributed to data acquisition and analysis. KX and YX were in charge of the writing. SL and SW participated in the implementation of the pilot trial. QY and QZ contributed to study design and manuscript revision. All authors approved the final version of the manuscript.

**Funding** This work was supported by the National Natural Science Foundation of China (grant number 71974170) and Medical Scientific Research Foundation of Zhejiang Province, China (grant number 2021ZH054). The funders have no influences on study methods, analysis, interpretation or publication of results.

**Competing interests** None declared.

**Patient and public involvement** Patients and/or the public were not involved in the design, or conduct, or reporting or dissemination plans of this research.

**Patient consent for publication** Not required.

**Provenance and peer review** Not commissioned; externally peer reviewed.

**ORCID iDs**
Kechen Xu http://orcid.org/0000-0001-5634-2534
Qian Yang http://orcid.org/0000-0002-5926-9309

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
