## [Reviewer comments · BMJ Open]

ARTICLE DETAILS

TITLE (PROVISIONAL)	The effect of positive emotion intervention during late pregnancy on improving colostrum secretion: a randomized control trial protocol
AUTHORS	Xu, Kechen; Xie, Ying; Han, Xiujun; Yu, Ying; Liu, Suqing; Wu, Suli; Yang, Qian; Zhang, Qi

VERSION 1 – REVIEW

REVIEWER	Berens, Pamela University of Texas
REVIEW RETURNED	17-Sep-2022

GENERAL COMMENTS	Overall the study is interesting and well designed. A few things could improve the document. 1) The authors describe checking PRL and IGF-1 levels. More details surrounding the timing of these levels relative to infant feeding are warranted (specifically the PRL level depends on when it is drawn relative to a feed / breast stimulation). Can the authors provide additional details regarding the timing of blood draws and what steps will be taken to mitigate the change in PRL levels relative to the blood draw itself. 2) Need further information regarding baseline frequency of breast stimulation for the study (will a mandatory minimum of at least 6-8 feedings / pumpings per day be required). Similarly, we need to know timing of the initial feed / breast stimulation relative to delivery as a potential confounder. 3) Also are involved hospitals Baby Friendly in their practices (skin to skin / rooming in, etc) That will help with interpreting external validity of results. 4) The authors provide detailed description of emotional testing - it would be ideal if they provided equally detailed description of infant feeding
--

REVIEWER	Schiøtz, Michaela The Danish Capital Region, Intersectoral Research Unit for Health Services
REVIEW RETURNED	09-Feb-2023

GENERAL COMMENTS	Thank you for the opportunity to review the manuscript "The effect of positive emotion intervention during late pregnancy on improving colostrum secretion: a randomized control trial protocol". The protocol presents a sound study with a relevant aim. I suggest that the following areas are addressed before publication: - In the abstract more information should be included about the intervention (in line 36)
--

	- In the outline of strengths and limitations it is stated that “Strong hospital health education on breastfeeding may obscure the effects of the experimental intervention and lead to negative results”. It is not clear what is meant by that, and it should be elaborated on. - In the section “Randomization and blinding” (line 182-189) more information about the randomization process should be given. This includes information about who will generate the randomization list, and who will carry out the randomization and give the participants information about which group they are allocated to. Also, information about when baseline data will be collected (before or after the randomization) should be provided. - In line 199-202 it is described how the participants are encouraged to fill out the questionnaires. Further information about how many times the participants are reminded/contacted and ethical considerations related to this should be provided.
--	--

VERSION 1 – AUTHOR RESPONSE

Reviewer: 1

Dr. Pamela Berens, University of Texas

Comments to the Author:

Overall the study is interesting and well designed. A few things could improve the document.

1) The authors describe checking PRL and IGF-1 levels. More details surrounding the timing of these levels relative to infant feeding are warranted (specifically the PRL level depends on when it is drawn relative to a feed / breast stimulation). Can the authors provide additional details regarding the timing of blood draws and what steps will be taken to mitigate the change in PRL levels relative to the blood draw itself.

Response: Thank you for the comments. We have added detailed descriptions on the timing of blood draws as shown, as the time between 7 and 8 a.m.

Method, Data collection and follow-up, Line 232-237:

“Blood samples will be collected twice, after they are admitted to the hospital approaching delivery and after delivery at 7 to 8 a.m. (Vaginal Delivery: the morning after one night; Cesarean section: the morning after two nights), and evaluated for prolactin and IGF-I to evaluate postpartum lactation ability. The blood draw will be conducted by nurses in the ward.”

Considering the following reasons.

First, according to the characteristics of PRL and IGF-1 rhythmic changes, hormone secretion is more stable between 7 a.m. and 8 a.m. [1,2]. The optimum sampling time for collecting serum prolactin and IGF-1 was around 8 in the morning [3,4]. Previous research exploring the effect of emotions on prolactin also draws blood at 8:00 a.m. [5].

Second, the hospital routinely requires blood draws at the time of antenatal admission and 1-2 days after delivery. In order to reduce the loss of pregnant women, we have combined the blood collection for the trial with the routine blood collection at the hospital.

Third, as our design was a randomized controlled trial (RCT), differences in between the two groups can be minimized, including the time of breastfeeding to blood draws.

Reference:

- [1] Salvador, J., Dieguez, C., & Scanlon, M. F. (1988). The Circadian Rhythms of Thyrotrophin and Prolactin Secretion. *Chronobiology International*, 5(1), 85–93.
- [2] Holden, J. P., Butzow, T. L., Laughlin, G. A., Ho, M., Morales, A. J., & Yen, S. C. (1995). Regulation of Insulin-Like Growth Factor Binding Protein-I During the 24-Hour Metabolic Clock and in Response to Hypoinsulinemia Induced by Fasting and Sandostatin in Normal Women. *Journal of the Society for Gynecologic Investigation*, 2(1), 38–44.
- [3] Hu Y, Ding Y, Yang M, Xiang Z. Serum prolactin levels across pregnancy and the establishment of reference intervals. *Clin Chem Lab Med*. 2018;56(5):838-842.
- [4] Zhu H, Xu Y, Gong F, et al. Reference ranges for serum insulin-like growth factor I (IGF-I) in healthy Chinese adults. *PLoS One*. 2017;12(10):e0185561. Published 2017 Oct 4.
- [5] Turner RA, Altemus M, Yip DN, et al. Effects of emotion on oxytocin, prolactin, and ACTH in women. *Stress Amst Neth*. 2002;5(4):269-276. doi:10.1080/1025389021000037586-1

2) Need further information regarding baseline frequency of breast stimulation for the study (will a mandatory minimum of at least 6-8 feedings / pumpings per day be required). Similarly, we need to know timing of the initial feed / breast stimulation relative to delivery as a potential confounder.

Response: Thank you for your advice. As we stated in the last comment, since our study design is RCT, these differences can be minimized in the two groups.

As you have mentioned, early frequent breast stimulation can contribute to lactation and breastfeeding behaviors. In our study, pregnant women will be routinely taught to breastfeed frequently when the milk rises or when the baby is hungry. Early initiation of breastfeeding stimulates the production of breast milk and also benefits the child [1,2]. Therefore, considering medical ethics, the involved hospitals and our team can only ask participants to start breastfeeding within the first hour after delivery.

Also, breastfeeding stimulation is the outcome indicator we use to assess on-demand feeding. So, we may not make a mandatory request on the baseline frequency of breastfeeding stimulation, but collect the average data during post-natal hospitalization.

References:

- [1] Ndirangu MN, Gatimu SM, Mwinyi HM, Kibiwott DC. Trends and factors associated with early initiation of breastfeeding in Namibia: analysis of the Demographic and Health Surveys 2000-2013. *BMC Pregnancy Childbirth*. 2018;18(1):171. Published 2018 May 16. doi:10.1186/s12884-018-1811-4
- [2] Abie BM, Goshu YA. Early initiation of breastfeeding and colostrum feeding among mothers of children aged less than 24 months in Debre Tabor, northwest Ethiopia: a cross-sectional study. *BMC Res Notes*. 2019;12(1):65. Published 2019 Jan 29. doi:10.1186/s13104-019-4094-6

3) Also are involved hospitals Baby Friendly in their practices (skin to skin / rooming in,etc) That will help with interpreting external validity of results.

Response: Thank you for the comment. We have added this point in the protocol and will collect measures taken by the involved Baby Friendly hospital in our study for better interpreting external validity of results.

Method, Data collection and follow-up, Line248-250:

“Generally, Baby-friendly hospitals will take measures such as early sucking and skin-to-skin contact within 1hr after delivery, limitation of bottle used, room-in, etc. We will also record and report routine measures taken in this Baby-Friendly hospital.”

4) The authors provide detailed description of emotional testing - it would be ideal if they provided equally detailed description of infant feeding such as an infant feeding log to assess how much of the infant's diet is breastmilk verses formula.

Response: Thank you for pointing this out. The recording of addition of formula milk during post-natal hospitalization is recorded by the nurse in this Baby-Friendly hospital. So, we can get more objective third-party record data in the nurse care record. We also added this point in the protocol.

Method, Evaluation and outcomes, Line212-215:

“Exclusive breastfeeding means that the baby is not given any liquid or solid food other than breast milk. This indicator will be included in the postnatal questionnaire and also be double-checked in the hospital care records of formula.”

Reviewer: 2

Dr. Michaela Schiøtz, The Danish Capital Region

Comments to the Author:

Thank you for the opportunity to review the manuscript “The effect of positive emotion intervention during late pregnancy on improving colostrum secretion: a randomized control trial protocol”. The protocol presents a sound study with a relevant aim. I suggest that the following areas are addressed before publication:

- In the abstract more information should be included about the intervention (in line 36)

Response: Thank you for pointing this out. We added the information on the intervention for the control group.

Methods and analysis, Line35-37:

“the intervention group will receive “Three Good Things” intervention while the control group will write about three things that come to mind first.”

- In the outline of strengths and limitations it is stated that “Strong hospital health education on breastfeeding may obscure the effects of the experimental intervention and lead to negative results”. It is not clear what is meant by that, and it should be elaborated on.

Response: Sorry for the expression of this sentence, we have corrected it. Currently, China's National Health Commission has called on hospitals at all levels to actively establish baby-friendly hospitals. And there are currently more than 7,000 ones in China. We want to explain that Baby-Friendly hospitals will be a trend in China. The settings in this study will offer evidence of real-world significance.

Strengths and limitations, Line59-61:

“We will promote breastfeeding-related behaviors in the postpartum hospitalization period in the setting of a Baby-Friendly hospital, in line with future trends in China.”

- In the section “Randomization and blinding” (line 182-189) more information about the randomization process should be given. This includes information about who will generate the randomization list, and who will carry out the randomization and give the participants information about which group they are allocated to. Also, information about when baseline data will be collected (before or after the randomization) should be provided.

Response: Thank you for your advice. We have revised it accordingly.

Method, Randomization and blinding, Line184-192:

“We will use stratified random grouping to divide the pregnant women into two groups according to the level of support they say they receive (high, average, none) from family members in the oral interview. Baseline data will be collected before the randomization. Pregnant women received on the same day will be randomly assigned to the intervention group and the control group according to the level of support (high, average, none). KCX will generate the randomization list, and YX will carry out the randomization and give the participants information about which group they are allocated to. Our trial will be a single-blinded trial of participants.”

- In line 199-202 it is described how the participants are encouraged to fill out the questionnaires. Further information about how many times the participants are reminded/contacted and ethical considerations related to this should be provided.

Response: Thank you for pointing this out. We have revised this part and added the ethical considerations.

Method, Intervention, Line204-208:

“When the participant is unwilling to take the initiative to fill in the questionnaire, we will ask for reasons and send encouraging messages (e.g. about the benefits of participating in the project). For ethical considerations of avoiding excessive disturbance, the participants will be regarded as loss-to-follow-up, after three non-responses.”